# Toward Model-Informed Precision Dosing for Remimazolam: A Population Pharmacokinetic–Pharmacodynamic Analysis

**DOI:** 10.3390/pharmaceutics16091122

**Published:** 2024-08-26

**Authors:** Yueting Chen, Cansheng Gong, Feng Liu, Zheng Jiao, Xiaochun Zheng

**Affiliations:** 1Department of Pharmacy, Shanghai Chest Hospital, Shanghai Jiao Tong University School of Medicine, Shanghai 200030, China; 3319091390@stu.cpu.edu.cn; 2Department of Anesthesiology, Shengli Clinical Medical College of Fujian Medical University, Fujian Provincial Hospital, Fuzhou University Affiliated Provincial Hospital, Fuzhou 350001, China; gongcansheng@163.com; 3Yichang Humanwell Pharmaceutical Co., Ltd., Wuhan 430075, China; liu_feng@renfu.com.cn; 4Fujian Provincial Key Laboratory of Emergency Medicine, Fujian Provincial Key Laboratory of Critical Care Medicine, Fujian Provincial Co-Constructed Laboratory of “Belt and Road”, Fuzhou 350001, China

**Keywords:** remimazolam, Monte Carlo simulation, population pharmacokinetics/pharmacodynamics models, individualized dosing regimens, web-based dashboard

## Abstract

Remimazolam, widely used for procedural sedation and general anesthesia, is a new ultra short-acting benzodiazepine for intravenous sedation and anesthesia. We aim to characterize the pharmacokinetics/pharmacodynamics (PK/PD) of remimazolam and its metabolite CNS 7054 in healthy Chinese volunteers using population analysis and suggest an optimal dosing regimen for sedation therapy. Data were collected from a single-center, placebo-controlled, randomized, and dose–escalation clinical pharmacology study. Forty-six healthy volunteers received a single infusion dose of remimazolam, while nine healthy subjects received a continuous infusion of remimazolam. A population PK/PD model was established and RxODE and Shiny in R were used to design the remimazolam dosing regimens. A three-compartment model best described the PK of remimazolam and a two-compartment model with one transit compartment was adopted for CNS 7054. The relationship between exposure and the bispectral index was best described using an effect compartment model with an inhibitory sigmoid model. Additionally, a web-based dashboard was developed to provide individualized dosing regimens, complemented by a graphical illustration of the PK/PD profiles of the proposed dosing regimen. The established population PK/PD model characterized the dose–exposure–response relationship of remimazolam well, which could be applied to optimize individual dosing regimens.

## 1. Introduction

Remimazolam, a novel benzodiazepine sedative, is recognized for its rapid onset, short duration of action, and minimal respiratory depression [1]. It produces sedative effects by selectively binding to γ-aminobutyric acid type A (γ-GABAA) receptors in the brain [2] and has been widely used for procedural sedation and general anesthesia [3,4].

Remimazolam is administrated intravenously (IV), with a plasma protein binding ratio of approximately 90%. In adults, it exhibits an apparent volume of distribution of 17.5 L. Remimazolam undergoes rapid metabolism primarily in the liver by carboxylesterase-1 (CES-1) to inactive metabolites (CNS 7054), which do not significantly involve cytochrome P450 (CYP) enzymes [5]. The terminal half-life is less than 1 h for remimazolam and 2.8 h for CNS 7054 [6,7]. CNS 7054 is primarily excreted through the kidneys, accounting for 70–90% of its elimination, whereas less than 1% of remimazolam is excreted [8].

In adults, the apparent clearance (CL) of CNS 7054 is relatively slow at 4.22 L/h, compared to remimazolam’s clearance of 70 L/h [6]. Additionally, a clinical study demonstrated that remimazolam, administered as a bolus in a dose range of 0.025–0.4 mg/kg and as a maintenance dose of 1–2 mg/kg/h, exhibited linear characteristics and did not show any accumulation. The sedative effects of anesthetic drugs, including remimazolam, are often assessed using the bispectral index (BIS) value via electroencephalogram (EEG) [9]. Therefore, the BIS value was monitored and considered a clinical measure of sedation for the PK/PD analysis.

Clinical studies conducted on healthy Chinese adult subjects for remimazolam employed intensive sampling, with its PK/PD characteristics elucidated through non-compartmental analysis (NCA) [7]. While NCA offers insight, the population PK/PD approach provides a comprehensive characterization of PK and PD profiles and designs dosing regimens based on target BIS values. Currently, the groundwork for utilizing population PK and PD parameters to develop a target-controlled infusion (TCI) system is insufficient. Furthermore, remimazolam is often used to sedate critically ill patients to maintain the BIS value within a target range [10], yet managing a precise level of sedation to avert instances of either oversedation or undersedation remains challenging. Despite numerous population PK/PD studies on remimazolam across different populations [11,12], an individualized dosing tool remains unavailable. These gaps highlight the urgent need for tailored research to develop personalized remimazolam treatment strategies.

Therefore, the aim of the study was to explore the PK/PD characteristics of remimazolam in Chinese subjects using population analysis. Additionally, we aimed to establish a population PK/PD model that could be utilized to develop a web-based tool for optimizing remimazolam dosage regimens.

## 2. Materials and Methods

### 2.1. Study Design

#### 2.1.1. Participants

Fifty-five healthy adult volunteers were recruited for the study. The inclusion criteria were as follows: (1) weight ≥ 50 kg; (2) age between 18 and 45 years; and (3) body mass index (BMI) ranging from 19 kg/m^2^ to 24 kg/m^2^. The exclusion criteria were as follows: (1) organ diseases or medical conditions that might hinder their participation in or completion of the study and (2) a score of 3 or 4 on the Mallampati scale. The specific study criteria were disclosed on the Chinese Clinical Trial Registry website (http://www.chictr.org.cn, accessed on 23 December 2019); ChiCTR1800015185 and ChiCTR1800015186). This study consists of a secondary analysis of the data [7].

The study was carried out at Peking University First Hospital, adhering to the principles outlined in the 2013 Declaration of Helsinki. The study protocol received approval from the Ethics Committee and written informed consent was acquired from all participants prior to their enrolment.

#### 2.1.2. Study Design

Forty-six of the volunteers received IV bolus doses of remimazolam (Human Well Co., Ltd., Wuhan, China), ranging from 0.025 mg/kg to 0.4 mg/kg, while nine received an IV bolus of 0.2 mg/kg/min followed by a maintenance dose of 1 mg/kg/h for 2 h. Detailed information is presented in Table 1. During the clinical trial, participants exclusively utilized remimazolam, adhering to a standardized diet regimen meticulously curated by the research team.

For subjects who received only IV bolus doses, venous blood samples were collected at 0, 1, 2, 3, 4, 5, 6, 8, 10, 12, 15, 20, 30, and 45 min and 1, 1.5, 2, 3, 4, 8, and 12 h post-dose. For subjects receiving both the bolus IV dose and maintenance dose, blood sampling was organized into two phases: during the initial 2-h infusion, samples were taken at 0, 1, 15, and 45 min and 1, 1.5, and 2 h after the start of the IV dose. Following the infusion period, samples continued to be drawn at 1, 2, 3, 4, 5, 6, 8, 10, 12, 15, 20, 30, and 45 min and 1, 1.5, 2, 3, 4, 6, and 10 h after the discontinuation of the remimazolam infusion.

Furthermore, the response of remimazolam was assessed by BIS values, which were continuously monitored through EEG (COVIDIEN, USA, BIS EEG VISTA) within 60 min post-dose for the IV bolus group and within 3 h post-dose for the subjects receiving both IV bolus and infusion.

#### 2.1.3. Bioassay

Quantification of remimazolam and its inactive metabolite, CNS7054, in the plasma was performed using validated high-performance liquid chromatography with tandem mass spectrometry detection (LC/MS/MS) [8]. Deuterium labeled d4—remimazolam and d4—CNS7054 were utilized as internal standards to determine the plasma concentrations of remimazolam and CNS7054, respectively.

The LC/MS/MS system used a 50 × 2 mm CAPCELL PAK MG C18 5-µm column (from SHISEIDO, Tokyo, Japan). Gradient elution was performed using two mobile phases: phase A consisted of 0.1% formic acid in a 10 mM ammonium formate aqueous solution and phase B was a mixture of 0.1% formic acid in a 2 mM ammonium formate with 95% acetonitrile aqueous solution. The nominal mass transitions from precursor ion to product ion were 439 to 362 for remimazolam and 425.1 to 407 for CNS 7054, respectively. The calibration range was 2–2000 ng/mL for both remimazolam and CNS7054. The coefficients of variation for inter- and intra-assay precision were ≤8.1% for remimazolam and ≤5.6% for CNS7054. The accuracy of the method ranged from 81.3 to 102% for remimazolam and from 91.3 to 106.6% for CNS7054.

### 2.2. Population Pharmacokinetics and Pharmacodynamics Modeling

A nonlinear, mixed-effect modeling program (NONMEM, version 7.5, ICON plc, Ellicott City, MD, USA) was used to develop the population PK/PD model. First-order conditional estimation with η-ε interaction method (FOCE-I) was used throughout the model development. Additionally, R (version 4.1.2) was utilized for statistical analysis and data visualization.

#### 2.2.1. Population Pharmacokinetics Modeling

Population PK models for remimazolam and CNS 7054 were established sequentially. Initially, we explored one-, two-, and three-compartment models with first-order elimination to determine the most appropriate PK structural model of remimazolam. Subsequently, CNS 7054 was linked to the remimazolam model by assuming that all remimazolam was converted into CNS 7054 in a first-order process. The conversion of remimazolam into CNS 7054 was evaluated with and without a time delay, described by either lag time or E-lang model [13]. After establishing the model, all PK parameters of remimazolam and CNS 7054 were simultaneously estimated to account for the interaction between the remimazolam and CNS 7054.

Between-subject variation (BSV) of PK parameters was characterized using exponential models, as shown in Equation (1), as follows:(1)$$P_{i}=\hat{P}\times exp(\eta^{P_{i}})$$
where $\eta^{P_{i}}$ is the BSV of the population parameters of the ith subject. $\eta^{P_{i}}$ is assumed to be normally distributed with a mean of 0 and variance of ω^2^.

Proportional, additive, and combined proportional and additive models were evaluated to describe residual unexplained variability (RUV).
(2)$$Y_{i,j}=F_{i,j}\times (1+\varepsilon_{1})$$
(3)$$Y_{i,j}=F_{i,j}+\varepsilon_{2}$$
(4)$$Y_{i,j}=F_{i,j}\times \left(1+\varepsilon_{1}\right)+\varepsilon_{2}$$
where $Y_{i,j}$ represents the ith subject of jth individual’s observed plasma concentration and $F_{i,j}$ is the ith subject of jth individual’s predicted plasma concentration. $\varepsilon_{1}$ and $\varepsilon_{2}$ are proportional and additive residual errors, respectively, and both are assumed to follow a normal distribution with a mean of 0 and variance of $\varepsilon^{2}$.

As the measurements of remimazolam and CNS 7054 were obtained from the same samples, the correlation of the RUV model between remimazolam and CNS 7054 was also examined.

The weight (WT) was integrated into the PK models via allometric scaling, as demonstrated in Equation (5), as follows:(5)$$P_{i}=\hat{P}\times {\left(\frac{{WT}_{i}}{60}\right)}^{\theta_{1}}$$
where $P_{i}$ denotes the parameter value of the ith subject while $\hat{P}$ represents the typical parameter value within the population. ${WT}_{i}$ represents the WT of the ith subject and $\theta_{1}$ serves as the exponent, with values set to 0.75 for CL and 1 for volume.

Other covariates including age, height, and sex were evaluated using a stepwise forward inclusion and backward elimination procedure. If the covariates are deemed significant, the objective function value (OFV) decreases by more than 3.84 (χ^2^ test, *p* < 0.05) during forward inclusion and increases by more than 6.63 (χ^2^ test, *p* < 0.01) during backward exclusion.

Continuous covariates, such as age and height, were normalized to the median population values using a power function (Equation (6)), as follows:(6)$$P_{i}=\hat{P}\times {\left(\frac{{Cov}_{i}}{{Cov}_{m}}\right)}^{\theta_{1}}$$
where $\theta_{1}$ represents the influence index of covariates on the parameters. ${Cov}_{i}$ is the covariate value of the ith subject and ${Cov}_{m}$ is the median value of the covariate.

Categorical covariates, such as sex, were integrated using a proportional model (Equation (7)), as follows:(7)$$P_{i}=\hat{P}\times {\theta_{2}}^{sex}$$
when the sex value was 0 or 1, indicating that the subject was male or female, respectively.

The selection of the model was determined by evaluating the OFV, Akaike information criterion (AIC) [14], Bayesian information criterion (BIC) [15], precision of parameter estimates, and pharmacological plausibility.

#### 2.2.2. Population PK/PD Modeling

After the population PK model was established, individual PK parameters were estimated using the empirical Bayes method and subsequently incorporated into the population PK/PD modeling [16].

An effect compartment with an inhibitory effect with or without the sigmoid index was evaluated to establish the relationship between remimazolam exposure and the BIS value according to previous reports [11]. The sigmoid $I_{max}$ model was shown as Equation (8), as follows:(8)$${BIS}_{i,j}={BIS}_{baseline,i}-\frac{I_{max,i}\times {CE}_{i,j}^{Hill}}{{IC}_{50,i}^{Hill}\times {CE}_{i,j}^{Hill}}$$
where ${BIS}_{baseline}$ is the value before administering remimazolam, $I_{max,i}$ is the maximum effect of remimazolam on BIS in the ith patient, ${IC}_{50,i}$ represents the concentration at half-maximum effect, ${CE}_{i,j}$ represents the concentration in the effect compartment in the ith subject at time jth, and Hill describes the shape of the relationship. The effective compartment concentrations were driven by a first-order rate constant ($k_{e0}$).

We assessed exponential and additive models to characterize BSV. For the RUV, we employed proportional, additive, and combined proportional-additive error models.

Then, age, height, and sex underwent forward inclusion (χ^2^ test, *p* < 0.05) and backward elimination (χ^2^ test, *p* < 0.01) during covariate screening.

The covariate model selection was informed by OFV, AIC [14], BIC [15], precision of parameter estimates, and pharmacological plausibility. After covariate screening, the PK and PD parameters were simultaneously estimated to account for the relationship between exposure and response [16,17].

#### 2.2.3. Model Evaluation

Goodness-of-fit (GOF) diagnosis plots were employed to examine the closeness between the observations and model predictions for the PK/PD model. In addition, visual predictive checks (VPCs) [18] were conducted by 1000 simulations to assess the prediction performance of the model. The 2.5th, 50th, and 97.5th percentiles of the simulated concentration distributions were assessed and compared with the observed data.

### 2.3. Dosing Regimen Design

To provide optimal individualized dosing regimens for critically ill patients [19,20], a web-based dashboard was developed using RxODE (version 1.1.5) and Shiny (version 1.7.1) in R to assess sedation levels. Anesthesiologists had the autonomy to individually tailor targeted BIS values in accordance with their professional assessments of critically ill patients. The optimal individual dosing regimens ensured that the patients reached a state of deep anesthesia within 3 min and maintained a consistent level throughout the administration period. Additionally, the BIS value–time curve for the optimal dosing regimen was generated using ggplot2 (version 3.2.1). Furthermore, to apply the dashboard in various clinical settings, a user-defined module was developed.

## 3. Results

### 3.1. Demographics

In total, 1113 PK observations for remimazolam, 1206 PK observations for CNS7054, and 1026 observed BIS from 55 volunteers were used to establish the PK/PD model. The study included 40 men and 15 women, with an age range of 19–43 years and a weight range of 52–75 kg. The demographic characteristics of the enrolled participants are summarized in Table 2. No significant differences were found among the subjects receiving different dosing regimens.

### 3.2. Population Pharmacokinetics and Pharmacodynamics Modeling

#### 3.2.1. Population Pharmacokinetics Modeling

A three-compartment model with first-order elimination was identified as the best structural model for remimazolam. The BSVs were incorporated into the CL, central volume (V1), peripheral volume (V3), and intercompartmental clearance (Q3).

CNS 7054, mainly originating from the direct metabolism of remimazolam, has been linked to the central compartment of the remimazolam PK model. To describe the transformation of the metabolite CNS 7054, a transit compartment was most appropriate to account for the delay between remimazolam and CNS 7054, as depicted in Figure 1. Similarly, a two-compartment model with first-order elimination was determined to be the best structural model for CNS 7054.

A proportional model was used to describe the RUV for remimazolam, while a combined proportional and additive model was utilized to characterize the RUV for CNS 7054. The additive error of CNS 7054 is 43.13 ng/mL, which is minimal compared to the concentrations of remimazolam, reaching a high of 30,000 ng/mL. A correlation of RUV between remimazolam and its metabolite CNS 7054 was also identified. No covariates were found to affect the PK of either remimazolam or CNS 7054.

The PK parameters for both remimazolam and its metabolite, CNS 7054, are summarized in Table 3. Compared to the inactive metabolite CNS 7054, remimazolam exhibited a significantly higher CL, of more than 21 times (CL: 1.36 vs. 0.0637 L/min), indicating a rapid transformation from the parent to the metabolite.

#### 3.2.2. Population PK/PD Modeling

We attempted to develop separate population PK models for remimazolam and CNS 7054, which revealed no significant differences in the model parameters compared to those obtained using a combined remimazolam and CNS 7054 PK modeling approach. Therefore, the PK of CNS 7054 exerts no significant influence on the PK profile of remimazolam; our approach focuses solely on the PK of remimazolam to construct the PK/PD model.

The sigmoid $I_{max}$ model best described the PK/PD relationship. BSV was incorporated into $I_{max}$ and ${IC}_{50}$ using an exponential model. The proportional model proved to be the most suitable for characterizing the RUV. No covariates were identified in the PD model. Table 4 displays the final model estimates of the remimazolam PD parameters.

#### 3.2.3. Model Evaluation

The population PK models of remimazolam and CNS 7054, as well as the population PK/PD model of remimazolam, were evaluated using GOF plots, as depicted in Figure 2, demonstrating the appropriateness of the model fit. The VPC plots are presented in Figure 3 and indicate that the established population PK/PD model effectively describes the PK and PD characteristics of remimazolam and CNS 7054.

### 3.3. Dosing Regimen Design

A web-based dashboard for individual dosing regimens can be accessed online (https://chen12yue-ting.shinyapps.io/Remimazolam/, accessed on 8 January 2024). Upon entering the desired BIS values, administration time, and patient demographics, the system can swiftly estimate the optimal personalized dosing regimens. These regimens include the induction and maintenance doses, with results available within seconds. Simultaneously, a graphical depiction of the BIS value–time curve for the selected optimal dosing regimen is displayed, offering a visual understanding of the expected pharmacodynamic effect.

The screenshots of this dashboard are displayed in Appendix A, depicting the case of a 40-year-old critically ill adult patient weighing 60 kg undergoing a 5-h administration. To ensure rapid attainment and maintenance of light sedation, the BIS values were set at 60–80. The optimal dosing regimen was a bolus injection of 0.1 mg/kg followed by an infusion rate of 0.6 mg/kg/h. The BIS value–time curve is also presented.

## 4. Discussion

This study is the first to analyze both the PK and PD profiles of remimazolam in Chinese people and develop a web-based application for recommending individually optimized dosage regimens of remimazolam. Three-compartment and effect compartment models were developed to describe the relationship between remimazolam exposure and BIS values. Furthermore, the optimal individual dosing regimens are recommended based on the established population PK/PD model using a web-based dashboard.

There are no significant differences in the PK of remimazolam compared to previous reports. Schüttler et al. [11] reported a CL of 1.14 L/min for white healthy subjects, which is consistent with our findings (1.2 L/min/60 kg). This is also consistent with a population PK study by Zhou et al. [21], who identified that race is not a clinically influential factor for PK (CL: 1.027 vs. 1.18 L/min/70 kg for African Americans vs. Asians and whites) across 11 phase I–III studies.

Similarly, the PD profiles of remimazolam were comparable to those reported in other studies. Zhou et al. [12] reported a similar ${IC}_{50}$ for white patients compared to that found in our study (504 vs. 496.4 ng/mL). However, for $I_{max}$, there were significant variations among studies. Previous studies [12,22] identified $I_{max}$ as 39.3 at a dose of 0.01–0.3 mg/kg and fixed at 73.3 for doses of 0.5 to 3 mg/kg/h, while our study reported $I_{max}$ of 54.1 for 0.025–0.4 mg/kg and 1 mg/kg/h. As for the rate constant for the effect compartment ($k_{e0}$), our study identified that $k_{e0}$ was faster than previous studies (1.09 min^−1^ vs. 0.141 min^−1^ [6] and 8.08 h^−1^ [12]). This difference may be attributed to the type of sample collected (veins vs. arteries) [22].

We found no clinically relevant covariate factors, which may be explained by the homogeneity of the healthy adult subjects. Previous studies [12,21] have identified sex, American Society of Anesthesiologists (ASA) status, and extracorporeal circulation (EC) as factors influencing the CL and central volume of remimazolam. In further studies, a more diverse population, including patients who are obese, pediatric, elderly, or have special conditions, should be investigated to identify the factors that influence the PK/PD of remimazolam.

It is important to precisely estimate the distribution volume of the central compartment (Vc), which can predict the maximal plasma concentration ($C_{max}$) and maximal efficacy. The established population PK model of remimazolam showed that the BSV of Vc was higher (56.8%) than that reported in a previous study (BSV of Vc: 28%) [22], which was closely correlated with the prediction $C_{max}$. Despite the studied population being homogeneous in its demographics, there may be other factors not considered that affect the BSV of Vc, such as genetic polymorphisms, disease states, and comorbidities. Further studies should be conducted to improve the accuracy of Vc prediction by identifying relevant covariates.

An established web-based dashboard was used for the determination of patient sedation regimens, based on studies conducted in healthy adult populations. Moreover, to accommodate patients’ diverse target BIS values for sedation, the upper and lower limits of the therapeutic range can be defined as needed in the user-defined module. It can be seen that the web-based dashboard provided an exemplary tool for recommending initial dosages for anesthesia or sedation. The PK and PD parameters of remimazolam could be user-defined. The dashboard could directly demonstrate the PK/PD profiles of remimazolam in patients. However, it also acknowledges the need for anesthesiologists to make real-time adjustments to accommodate procedural variabilities due to the large variances among patients during the anesthesia or sedation process. While the web-based dashboard currently lacks the functionality to estimate Bayesian PK parameters from plasma concentration data, this feature is envisioned to be incorporated in future iterations of the research as it matures.

This study had several limitations. First, the population PK/PD model was established in healthy adult subjects who were relatively homogenous. Therefore, the model may not be extrapolated to patients with large differences from healthy subjects. Additionally, the study subjects only received remimazolam and did not explore the synergistic effect of co-therapy with benzodiazepines and opioids, which requires further investigation in future studies. Finally, the web-based dashboard currently has limited usage in clinical practice due to its incompletely refined features.

## 5. Conclusions

In conclusion, our study demonstrated that there were no significant effects of covariates on the PK and PD characteristics of remimazolam. The established dose–exposure–response relationship of remimazolam aptly delineated its PK and PD profiles, facilitating rapid onset and recovery in subjects. Additionally, the established population PK/PD model of remimazolam can be employed to optimize individualized therapy using a web-based dashboard.

## Figures and Tables

**Figure 1 pharmaceutics-16-01122-f001:**
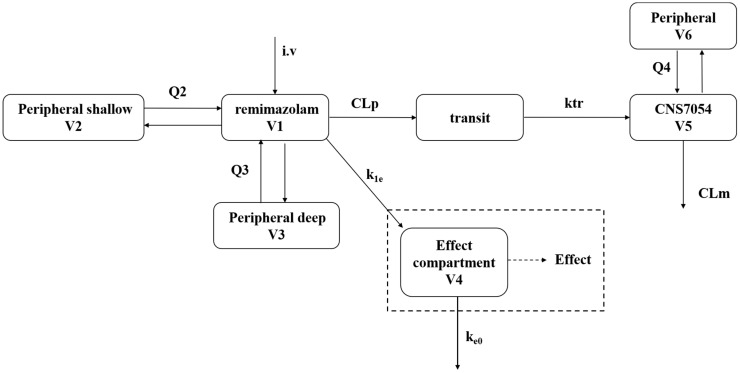
**Schematic of population PK and PD models of remimazolam.** Combined pharmacokinetic/pharmacodynamic model, consisting of a three-compartment model for the parent drug remimazolam, a two-compartment model for the metabolite CNS 7054, a transit compartment for the formation of the metabolite, and an effect compartment model with an inhibitory sigmoid Emax model. V_1,5_, central volumes of distribution; V_2,3,6_, peripheral volumes of distribution; CL_p_, CL_m_, and k_e0_, elimination clearances; Q_2,3,4_ and k_1e_, intercompartmental clearances; and k_tr_, transit rate constant.

**Figure 2 pharmaceutics-16-01122-f002:**
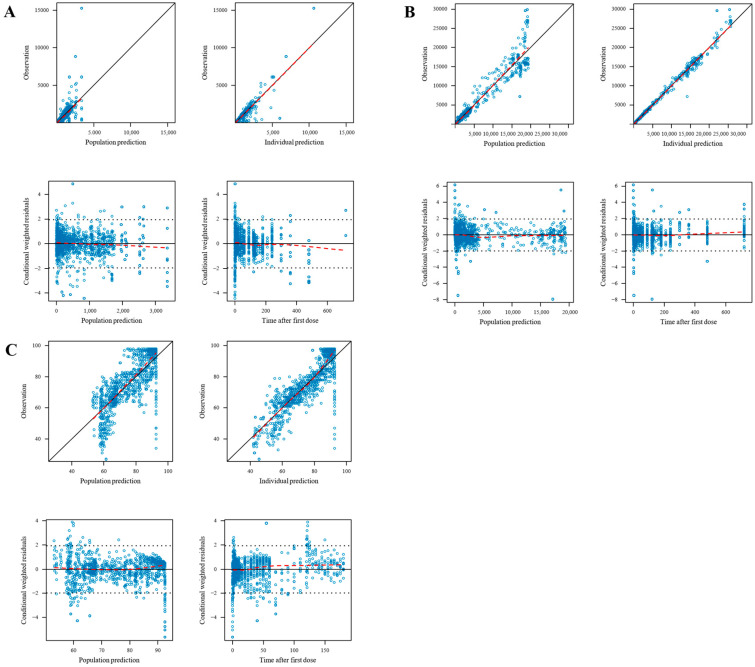
**Goodness of fit plots of the population PK/PD model.** (**A**). The population PK model for remimazolam; (**B**). The population PK model for CNS 7054; (**C**). The population PD model for remimazolam.

**Figure 3 pharmaceutics-16-01122-f003:**
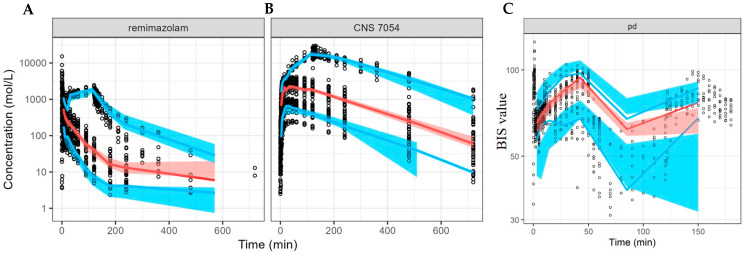
Visual predictive check plots of the population PK/PD model. (**A**). Logarithmic scale for the population PK model of remimazolam; (**B**). Logarithmic scale for the population PK model of CNS 7054; (**C**). Normal scale for the population PD model. Blue solid lines represent the 10th, 50th, and 90th percentiles of the observations; Red dashed lines represent the 5th, 50th, and 95th percentiles of the simulated values.

**Table 1 pharmaceutics-16-01122-t001:** Study design of included subjects in the PopPK and PopPK/PD analyses.

Study Number	Dose Regimen and Number of Subjects	Patient Population with PK/PD Sampling	PK Sampling	PD Sampling
1	IV bolus: 0.025 mg/kg: 3; 0.05 mg/kg: 3; 0.075 mg/kg: 8; 0.1 mg/kg: 8; 0.2 mg/kg: 8; 0.3 mg/kg: 10; 0.4 mg/kg: 8;	46 Healthy Adult subjects	0, 1, 2, 3, 4, 5, 6, 8, 10, 12, 15, 20, 30, 45 min, and 1, 1.5, 2, 3, 4, 8, 12 h post dose	0, 1, 2, 5, 10, 20, 30, 40, 50, 60 min post dose
2	IV bolus: 0.2 mg/kg/min for 1 min; IV infusion: 1 mg/kg/h for 2 h;	9 Healthy Adult subjects	Part I: 0, 1, 15, 45 min, and 1, 1.5, 2 h after IV dose initiation for 2 h infusion; Part II: 1, 2, 3, 4, 5, 6, 8, 10, 12, 15, 20, 30, 45 min, 1, 1.5, 2, 3, 4, 6, 10 h post dose.	Part I: 0, 1, 2, 3, 4, 5, 6, 7, 8, 9, 10, 12, 15, 20, 25, 30, 35, 40, 45, 50, 55, 60, 70, 80, 90, 100, 110 min after IV dose initiation for 2 h infusion; Part II: 1, 2, 5, 10, 15, 20, 25, 30, 35, 40, 45, 50, 55, 60 min post dose.

Abbreviations: PK, pharmacokinetics; PD, pharmacodynamics; IV, intravenous injection.

**Table 2 pharmaceutics-16-01122-t002:** Demographic characteristics of the subjects.

Characteristic	Multi-Dose (*N* = 9) Median (Min, Max)	Single-Dose (*N* = 46) Median (Min, Max)	Total (*N* = 55) Median (Min, Max)
Sex (male/female)	7/2		
Age (years)	28 (23, 43)	28 (19, 41)	28 (19, 43)
BMI (kg/m^2^)	22.4 (19.5, 23.8)	22.3 (19.3, 24)	22.4 (19.3, 24)
Height (cm)	166 (151, 176)	169 (154, 185)	167.5 (151, 185)
Weight (kg)	61.8 (52.5, 67)	63.2 (52,75)	62.5 (52, 75)

Abbreviations: BMI, Body Mass Index.

**Table 3 pharmaceutics-16-01122-t003:** Final popPK model parameter estimation of remimazolam and CNS 7054.

Parameters	Estimate (RSE%)	Shrinkage (%)	Parameters	Estimate (RSE%)	Shrinkage (%)
**Typical value**					
CL for remimazolam (L/min)	1.21 (3%)	/	CL for CNS7054 (L/min)	0.0637 (3%)	/
V1 for remimazolam (L)	16 (8%)	/	V5 for CNS7054 (L)	3.72 (5%)	/
V2 for remimazolam (L)	22.6 (3%)	/	Q4 for CNS7054 (L/min)	0.166 (5%)	/
Q2 for remimazolam (L/min)	2.61 (4%)	/	V6 for CNS7054 (L)	5.15 (4%)	/
Q3 for remimazolam (L/min)	0.227 (14%)	/			
V3 for remimazolam (L)	23.5 (6%)	/			
Ktr for remimazolam (1/min)	0.447 (8%)	/			
**Inter-individual variation (IIV)**					
IIV_CL for remimazolam (%)	20 (10%)	2	IIV_CL for CNS7054 (%)	22.1 (10%)	2
IIV_V1 for remimazolam (%)	55 (10%)	3	IIV_V5 for CNS7054 (%)	30.1 (12%)	7
IIV_Q2 for remimazolam (%)	24.3 (26%)	30	IIV_V6 for CNS7054 (%)	17.5 (15%)	22
IIV_V2 for remimazolam (%)	32.1 (13%)	9			
IIV_Ktr for remimazolam (%)	40.1 (14%)	17			
**Residual unexplained variability (RUV)**					
prop RUV for remimazolam (%)	23.2 (1%)	7	prop RUV for CNS7054 (%)	6.4 (0%)	14
COR (remimazolam _CNS7054)	0.0066	/	add RUV for CNS 7054	43.13 (8%)	8

Abbreviations: CL, clearance; V1, central volume; V2, peripheral volume; V3, peripheral volume; Q2, intercompartmental clearance; Q3, intercompartmental clearance; Ktr, transit rate constant; V5, central volume for CNS 7054; V6, peripheral volume for CNS 7054; Q4, intercompartmental clearance for CNS 7054. RSE, residual standard error.

**Table 4 pharmaceutics-16-01122-t004:** Final popPD model parameter estimation of remimazolam.

Parameters	Estimate (RSE%)	Shrinkage (%)
**Typical value**		
$I_{max}$	54.5 (6%)	/
${IC}_{50}$ (ng/mL)	504 (9%)	/
BIS_baseline	92.5 (1%)	/
$k_{e0}$ (min^−1^)	1.38 (26%)	/
Hill coefficient	1.44 (10%)	/
**Inter-individual variation (IIV)**		
IIV_$I_{max}$ (%)	27.2 (16%)	28
IIV_HILL (%)	48.8 (13%)	17
**Residual unexplained variability (RUV)**		
prop RUV pd (%)	11.2 (2.4%)	4

Abbreviations: $k_{e0}$, elimination rate constant; RSE, residual standard error.

## Data Availability

The original contributions presented in the study are included in the article/Appendix A; further inquiries can be directed to the corresponding author.

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
