# Peer review of "Toward Model-Informed Precision Dosing for Remimazolam: A Population Pharmacokinetic–Pharmacodynamic Analysis"

_pharmaceutics, 2024, doi:10.3390/pharmaceutics16091122_

Round 1

Reviewer 1 Report

Comments and Suggestions for Authors

The authors measured plasma concentrations of remimazolam and its metabolite CNS7054 in healthy subjects, which is widely used for sedation and anesthesia, and constructed a PPK model. They also constructed a PD model that shows the relationship between plasma remimazolam and BIS. Based on these PK/PD models, they provided a web tool to easily set clinical doses. The manuscript is generally well written, but I have some comments.

In abstract, BIS should be spell out to bispectral index.

Section 2.2.1.

Regarding CNS7054, have you considered a model that links to the peripheral compartment?

Section 3.2.1.

Does the PK of CNS7054 affect the PK of remimazolam, i.e., CLp? Since CNS7054 is an inactive metabolite and the PK of CNS7054 has a larger delay than the PK of remimazolam, wouldn't it be sufficient to optimize only the PK of remimazolam (excluding CNS7054 model)?

Section 5.

I think that it is useful to construct a PPK model for remimazolam in this study. However, as a result, no covariates affecting the PK of remimazolam and CNS7054 were found. Therefore, although a web-based dashboard based on the PK/PD model was provided, it can only show the dose and the population average response. To optimize individual treatment, it is necessary to measure the patient's plasma concentration and estimate individual posthoc parameters. It seems unrealistic to measure remimazolam concentrations and individualize in clinical practice. Also, does this dashboard have a function to Bayesian estimate PK parameters from plasma concentrations?

From another perspective, it may be possible to optimize the dose by PK parameters and posthoc PD parameter values from individual PD index values. However, there is a risk of large error in estimation from measurements where the initial PD change after administration is large. In areas where the PD effect is saturated, the significance of using this tool may be diminished.

Please discuss how this tool can be used.

Author Response

The authors measured plasma concentrations of remimazolam and its metabolite CNS7054 in healthy subjects, which is widely used for sedation and anesthesia, and constructed a PPK model. They also constructed a PD model that shows the relationship between plasma remimazolam and BIS. Based on these PK/PD models, they provided a web tool to easily set clinical doses. The manuscript is generally well written, but I have some comments.

  1. In abstract, BIS should be spell out to bispectral index.

Response: Thank you for your advice. We have screened the text and revised the manuscript as suggested.

  1. Section 2.2.1. 

Regarding CNS7054, have you considered a model that links to the peripheral compartment?

Response: Thank you for your advice. We have revised the manuscript as suggested.

Page 1, line 41:

Remimazolam undergoes rapid metabolism primarily in the liver by carboxylesterase-1 (CES-1) to inactive metabolites (CNS 7054) ……

Page 6, line 233:

CNS 7054, mainly originating from the direct metabolism of remimazolam, has been linked to the central compartment of the remimazolam PK model.

  1. Section 3.2.1. 

Does the PK of CNS7054 affect the PK of remimazolam, i.e., CLp? Since CNS7054 is an inactive metabolite and the PK of CNS7054 has a larger delay than the PK of remimazolam, wouldn't it be sufficient to optimize only the PK of remimazolam (excluding CNS7054 model)?

Response: Thank you for your advice. We have revised the manuscript as suggested.

Page 8, line 263:

We attempted to develop separate population PK models for remimazolam and CNS 7054, which revealed no significant differences in the model parameters compared to those obtained using a combined remimazolam and CNS 7054 PK modeling approach. Therefore, the PK of CNS 7054 exerts no significant influence on the PK profile of remimazolam, and our approach focuses solely on the PK of remimazolam to construct the PK/PD model.

  1. Section 5.

I think that it is useful to construct a PPK model for remimazolam in this study. However, as a result, no covariates affecting the PK of remimazolam and CNS7054 were found. Therefore, although a web-based dashboard based on the PK/PD model was provided, it can only show the dose and the population average response. To optimize individual treatment, it is necessary to measure the patient's plasma concentration and estimate individual posthoc parameters. It seems unrealistic to measure remimazolam concentrations and individualize in clinical practice. Also, does this dashboard have a function to Bayesian estimate PK parameters from plasma concentrations?

Response: Thank you for your advice. We have revised the manuscript as suggested.

Page 10, line 352:

While the web-based dashboard currently lacks the functionality to estimate Bayesian PK parameters from plasma concentration data, this feature is envisioned to be incorporated in future iterations of the research as it matures.

Page 10, line 358:

Finally, the web-based dashboard currently has limited usage in clinical practice due to its incompletely refined features.

  1. From another perspective, it may be possible to optimize the dose by PK parameters and posthoc PD parameter values from individual PD index values. However, there is a risk of large error in estimation from measurements where the initial PD change after administration is large. In areas where the PD effect is saturated, the significance of using this tool may be diminished.

Response: Thank you for your advice.

Page 10, line 346:

It can be seen that the web-based dashboard provided an exemplary tool for recommending initial dosages for anesthesia or sedation.

Page 10, line 350:

However, it also acknowledges the need for anesthesiologists to make real-time adjustments to accommodate procedural variabilities, due to the large variances among patients during the anesthesia or sedation process.

  1. Please discuss how this tool can be used.

Response: Thank you for your advice. We have revised the manuscript as suggested.

Page 10, line 346:

It can be seen that the web-based dashboard provided an exemplary tool for recommending initial dosages for anesthesia or sedation. The PK and PD parameters of remimazolam could be user-defined. And the dashboard could directly demonstrate the PK/PD profiles of remimazolam in patients. However, it also acknowledges the need for anesthesiologists to make real-time adjustments to accommodate procedural variabilities, due to the large variances among patients during the anesthesia or sedation process. While the web-based dashboard currently lacks the functionality to estimate Bayesian PK parameters from plasma concentration data, this feature is envisioned to be incorporated in future iterations of the research as it matures.

Reviewer 2 Report

Comments and Suggestions for Authors

The manuscript describes development of PK/PD model using non-linear mixed effects analysis to describe the PKPD of remimazolam and its metabolite in healthy participants of Chinese ethnicity. The authors also describe development of a web-based dashboard to support dose individualization.

The paper is well-written in that it flows in a logical progression for various aspects of model development and eventual application.

Major comments

1.       It is not clear why the authors used an allometric approach for weight as a covariate on CL and V, but then used a power function for other covariates. It seems a power function should have been used for weight as well. Also, the covariate model should not include BMI as a potential covariate if weight is used, since BMI depends on weight. Either use BMI alone, or weight and height separately during covariate model development. The authors may have done this, but should explain if they did.

2.       The authors’ need to comment on possible reasons for the high IIV for V1 compared to published models, and in view of their comments that the population was homogeneous in its demographics.

Minor comments

1.       Methods, page 3 in paragraph following Table 1, specify Part I and Part II before the listing of the timepoints for each part. Part II is also misspelled in the table.

2.       Line 105, page 3: delete “benzenesulfonate” since the free base is what is measured, not the salt.

3.       Line 201 on page 5, it is not clear what “user defined” means for the BIS values used in the model.

4.       Results, Figure 1 caption title: remove “Goodness of fit plots of population PK/PD model.”

5.       Results, Page 6 and Table 3: the additive error for the metabolite is quite high. The authors need to comment on possible reasons for this.

6.       Conclusions: Page 10, line 332: “There was (were) no significant effect of ??? on the PK and PD characteristics of remimazolam.” What effects?

Author Response

Comments to the Author

The manuscript describes development of PK/PD model using non-linear mixed effects analysis to describe the PKPD of remimazolam and its metabolite in healthy participants of Chinese ethnicity. The authors also describe development of a web-based dashboard to support dose individualization. The paper is well-written in that it flows in a logical progression for various aspects of model development and eventual application.

Major comments

  1. It is not clear why the authors used an allometric approach for weight as a covariate on CL and V, but then used a power function for other covariates. It seems a power function should have been used for weight as well. Also, the covariate model should not include BMI as a potential covariate if weight is used, since BMI depends on weight. Either use BMI alone, or weight and height separately during covariate model development. The authors may have done this, but should explain if they did.

Response: Thank you for your advice. We had excluded the BMI as a potential covariate.

Weight was incorporated into the population PK model due to dosing based on weight, which reflects the physiological basis for this method. Therefore, we used an allometric approach for weight. For other covariates, we used a power function to identify for significance effects.

  1. The authors’ need to comment on possible reasons for the high IIV for V1 compared to published models, and in view of their comments that the population was homogeneous in its demographics.

Response: Thank you for your advice. We have revised the manuscript as suggested.

Page 10, line 338:

Despite the studied population being homogeneous in its demographics, there may be other factors not considered that affect the BSV of Vc, such as genetic polymorphisms, disease states, and comorbidities.

Minor comments 

  1. Methods, page 3 in paragraph following Table 1, specify Part I and Part II before the listing of the timepoints for each part. Part II is also misspelled in the table.

Response: Thank you for your advice. We have revised the manuscript as suggested.

Page 3, line 98:

For subjects receiving both the bolus IV dose and maintenance dose, blood sampling was organized into two phases: during the initial 2-hour infusion, samples were taken at 0, 1, 15, and 45 min, and 1, 1.5, and 2 h after the start of the IV dose. Following the infusion period, samples continued to be drawn at 1, 2, 3, 4, 5, 6, 8, 10, 12, 15, 20, 30, and 45 min, and 1, 1.5, 2, 3, 4, 6, and 10 h after the discontinuation of the remimazolam infusion.

  1. Line 105, page 3: delete “benzenesulfonate” since the free base is what is measured, not the salt.

Response: Thank you for your advice. We have revised the manuscript as suggested.

  1. Line 201 on page 5, it is not clear what “user defined” means for the BIS values used in the model.

Response: Thank you for your advice. We have revised the manuscript as suggested.

Page 5, line 210:

Anesthesiologists had the autonomy to individually tailor targeted BIS values in accordance with their professional assessments of critically ill patients.

  1. Results, Figure 1 caption title: remove “Goodness of fit plots of population PK/PD model.”

Response: Thank you for your advice. We have revised the manuscript as suggested.

  1. Results, Page 6 and Table 3: the additive error for the metabolite is quite high. The authors need to comment on possible reasons for this.

Response: Thank you for your advice. We have revised the manuscript as suggested.

Page 7, line 249:

The additive error of CNS 7054 is 43.13 ng/mL, which is minimal compared to the concentrations of remimazolam, reaching a high of 30000 ng/mL.

  1. Conclusions: Page 10, line 332: “There was (were) no significant effect of ??? on the PK and PD characteristics of remimazolam.” What effects?

Response: Thank you for your advice. We have revised the manuscript as suggested.

Page 11, line 364:

In conclusion, our study demonstrated that there were no significant effects of covariates on the PK and PD characteristics of remimazolam.

Reviewer 3 Report

Comments and Suggestions for Authors

1. This article characterizes the pharmacokinetics/pharmacodynamics (PK/PD) of remimazolam and its metabolite CNS 7054 in healthy Chinese volunteers through population analysis. However, there are several questions that need to be answered by the authors.

The authors should consider the influence of gender during the study,

The authors should consider the influence of diet and whether there are other medications during the study

The authors should list the LC/MS/MS detection conditions of remimazolam

The horizontal and vertical coordinates of Figure 2 are not clear and need to be redrawn

Comments on the Quality of English Language

English language needs to be polished.

Author Response

This article characterizes the pharmacokinetics/pharmacodynamics (PK/PD) of remimazolam and its metabolite CNS 7054 in healthy Chinese volunteers through population analysis. However, there are several questions that need to be answered by the authors.

  1. The authors should consider the influence of gender during the study,

Response: Thank you for your advice. We have revised the manuscript as suggested.

Page 5, line 220:

The study included 40 men and 15 women……

Page 4, line 162:

Other covariates including age, height, and sex, were evaluated using a stepwise forward inclusion and backward elimination procedure.

Page 7, line 251:

No covariates were found to affect PK of either remimazolam or CNS 7054.

  1. The authors should consider the influence of diet and whether there are other medications during the study.

Response: Thank you for your advice. We have revised the manuscript as suggested.

Page 2, line 90:

During the clinical trial, participants exclusively utilized remimazolam, adhering to a standardized diet regimen meticulously curated by the research team.

  1. The authors should list the LC/MS/MS detection conditions of remimazolam.

Response: Thank you for your advice. We have revised the manuscript as suggested.

Page 3, line 113:

The LC/MS/MS system used a 50 × 2mm CAPCELL PAK MG C18 5-µm column (from SHISEIDO). Gradient elution was performed using two mobile phases: phase A consisted of 0.1% formic acid in a 10 mM ammonium formate aqueous solution, and phase B was a mixture of 0.1% formic acid in a 2 mM ammonium formate with 95% acetonitrile aqueous solution. The nominal mass transitions from precursor ion to product ion were 439 to 362 for remimazolam, and 425.1 to 407 for CNS 7054, respectively.

  1. The horizontal and vertical coordinates of Figure 2 are not clear and need to be redrawn

Response: Thank you for your advice. We have revised the manuscript as suggested.

Round 2

Reviewer 1 Report

Comments and Suggestions for Authors

The authors measured plasma concentrations of remimazolam and its metabolite CNS7054 in healthy subjects, which is widely used for sedation and anesthesia, and constructed a PPK model. They also constructed a PD model that shows the relationship between plasma remimazolam and BIS. Based on these PK/PD models, they provided a web tool to easily set clinical doses.

The authors added the reasonable responses.

Reviewer 3 Report

Comments and Suggestions for Authors

The author answered the questions raised well. This revision is acceptable.

Comments on the Quality of English Language

 Minor editing of English language required